# Andrographolide Promotes Uptake of Glucose and GLUT4 Transport through the PKC Pathway in L6 Cells

**DOI:** 10.3390/ph15111346

**Published:** 2022-10-31

**Authors:** Jingya Liao, Ziwei Yang, Yanhong Yao, Xinzhou Yang, Jinhua Shen, Ping Zhao

**Affiliations:** 1Institute for Medical Biology & Hubei Provincial Key Laboratory for Protection and Application of Special Plants in the Wuling Area of China, College of Life Sciences, South-Central Minzu University, Wuhan 430074, China; 2School of Pharmaceutical Sciences, South-Central Minzu University, Min-Zu Road, Wuhan 430074, China

**Keywords:** andrographolide, GLUT4, Ca^2+^, L6 cells, type 2 diabetes

## Abstract

Glucose transporter 4 (GLUT4) is a membrane protein that regulates blood glucose balance and is closely related to type 2 diabetes. Andrographolide (AND) is a diterpene lactone extracted from herbal medicine Andrographis paniculata, which has a variety of biological activities. In this study, the antidiabetic effect of AND in L6 cells and its mechanism were investigated. The uptake of glucose of L6 cells was detected by a glucose assay kit. The expression of GLUT4 and phosphorylation of protein kinase B (PKB/Akt), AMP-dependent protein kinase (AMPK), and protein kinase C (PKC) were detected by Western blot. At the same time, the intracellular Ca^2+^ levels and GLUT4 translocation in myc-GLUT4-mOrange-L6 cells were detected by confocal laser scanning microscopy. The results showed that AND enhanced the uptake of glucose, GLUT4 expression and fusion with plasma membrane in L6 cells. Meanwhile, AND also significantly activated the phosphorylation of AMPK and PKC and increased the concentration of intracellular Ca^2+^. AND-induced GLUT4 expression was significantly inhibited by a PKC inhibitor (Gö6983). In addition, in the case of 0 mM extracellular Ca^2+^ and 0 mM extracellular Ca^2+^ + 10 μM BAPTA-AM (intracellular Ca^2+^ chelator), AND induced the translocation of GLUT4, and the uptake of glucose was significantly inhibited. Therefore, we concluded that AND promoted the expression of GLUT4 and its fusion with plasma membrane in L6 cells through PKC pathways in a Ca^2+^—dependent manner, thereby increasing the uptake of glucose.

## 1. Introduction

According to the international diabetes federation (IDF), by 2045, the global prevalence of diabetes will be increased from 9.3% in 2019 (463 million) to 10.9% (700 million people), while over 90% of patients with diabetes mellitus have type 2 diabetes mellitus (T2DM) [1]. T2DM is a multi-etiological lifelong metabolic disease characterized by chronic hyperglycemia accompanied by disturbances in the metabolism of sugars, fats, and proteins due to defective insulin secretion or function [2,3,4,5]. Long-term metabolic abnormalities lead to chronic damage and dysfunction of various tissues, especially the eyes, kidneys, and nerves, causing a series of complications such as diabetic nephropathy and cardiovascular and cerebrovascular complications of diabetes, which seriously threaten the life safety of patients [6,7,8]. Therefore, maintaining normal glucose metabolism is essential for health.

Glucose transport protein 4 (GLUT4) is a member of the family of glucose transporters that mainly exists in skeletal muscle, adipose tissue, and the heart. It is a membrane protein that insulin regulates to lower blood glucose and has a 12-fold transmembrane domain [9]. GLUT4 is responsible for extracellular glucose transport and plays a central role in whole-body glucose metabolism, which is closely related to T2DM [10,11]. GLUT4 storage vesicles (GSV) are transportable vesicle carriers that exist in fat and skeletal muscle cells [12,13]. GLUT4 translocation is mainly induced by insulin-dependent and insulin-independent pathways [14,15]. Studies have shown that the phosphorylation of protein kinase B (Akt/PKB) and its upstream and downstream proteins, as well as the activation of AMP-activated protein kinase (AMPK), can promote the translocation of GLUT4, so as to promote the uptake of glucose by skeletal muscle [16,17,18,19]. Other studies have shown that the phosphorylation of the atypical protein kinase C (aPKC) pathway plays an important role in GLUT4 translocation and uptake of glucose [20,21]. The increase of Ca^2+^ in cytosol mediates muscle contraction, which in turn induces the transfer of GLUT4 from inside the cell to the cell surface to increase glucose transport [22,23]. These reports provided reference for studying the mechanism of different hypoglycemic drugs.

Some studies in the past have shown that many active ingredients of herbal medicine have significant anti-diabetes effects, such as panax notoginseng total saponins [24], cassia seed [25], neferine [26], and berberine [27]. Andrographolide (AND) is a diterpenoid lactone compound extracted from the herb named Andrographis paniculata [28], which has a variety of biological effects, including anti-inflammatory, antioxidant, and anti-tumor effects [29,30]. Regarding the therapeutic effect of AND on diabetes, previous studies have shown that AND has a hypoglycemic effect on diabetic mice induced by streptozotocin (STZ) [31] and can also improve lipid metabolism and glucose utilization in obese mice induced by a high-fat diet [32]. AND can reduce hyperglycemia-mediated renal oxidative stress and inflammation and improve diabetic nephropathy [33]. In our study, we confirmed that AND promotes the expression of GLUT4 through the PKC pathway, thereby promoting GLUT4 fusion into the plasma membrane and the uptake of glucose. We also demonstrated that AND-induced GLUT4 fusion into the plasma membrane and the uptake of glucose were dependent on intracellular Ca^2+^. In conclusion, these studies further provide a reference for the anti-diabetic molecular mechanism of AND with GLUT4 as the target.

## 2. Results

### 2.1. AND Was the Major Component in AP-EtOAc

The homogeneity was compared with representative LC-ESIMS chromatograms between AP-EtOAc and AND in Figure 1. We found that AP-EtOAc and AND had the same retention time. AND was found to be the main active ingredient in AP-EtOAc with content of about 22.71%. Therefore, we hypothesized that AND was the basis of the GLUT4-targeting anti-diabetic effect of AP-EtOAc. Next, we investigated the anti-diabetic activity of AND targeting GLUT4.

### 2.2. AP-EtOAc Promoted Uptake of Glucose and Enhances GLUT4 Transport

In this study, we first analyzed the effect of AP-EtOAc on the uptake of glucose in L6 cells by using a glucose (GLU-OX) assay kit (glucose oxidase method). We supposed that AP EtOAc could increase the uptake of glucose in both insulin resistance (IR) and normal L6 cells, while 100 nM insulin only induced the increase of glucose uptake of non-insulin-resistant cells, as shown in Figure 2A,B. In the non-insulin-resistance model, after adding drugs to incubate L6 cells, the effect of 60 μg/mL AP-EtOAc on the uptake of glucose was significantly higher than that in the control group, and the uptake of glucose effect of 90 μg/mL AP-EtOAc was comparable to that of insulin. By contrast, the effect of the drug on the promotion of the uptake of glucose was dose-dependent. Next, in order to study whether AP-EtOAc could promote the uptake of glucose by enhancing the transport of GLUT4 in cells, we observed the changes of GLUT4 levels in cells by laser confocal microscopy. The results showed that in the myc-GLUT4-mOrange-L6 cells treated with AP-EtOAc at 60 μg/mL within 30 min, the level of red fluorescent protein (GLUT4-mOrange) in cells increased rapidly by about two-fold (Figure 2C,D). 

### 2.3. AP-EtOAc Induced GLUT4 Protein Expression Levels and Fusion into Plasma Membrane in L6 Cells

After AP-EtOAc induced the transport of GLUT4 in myc-GLUT4-mOrange-L6 cells by a confocal laser scanning microscope (CLSM), we considered whether AP-EtOAc could also promote GLUT4 fusion to the plasma membrane. Therefore, in order to verify this conclusion, we performed an immunofluorescence experiment to detect red fluorescence (GLUT4-mOrange) and green fluorescence (FITC-myc). As a result, we found that both 60 μg/mL AP-EtOAc and 100 nM insulin could promote the fusion of GLUT4 with the plasma membrane (Figure 3A). We counted the cell surface FITC fluorescence to quantify this effect and found that the proportion of FITC fluorescent cells in GLUT4-mOrange positive cells under AP-EtOAc and insulin was 70% and 85%, respectively (Figure 3B). Then, we examined the expression levels of the GLUT4 protein in L6 cells. After adding 60 μg/mL and 90 μg/mL AP-EtOAc within 30 min, the expression of GLUT4 protein increased by approximately 0.9-fold and 1.3-fold, respectively (Figure 3C). We concluded that AP-EtOAc promoted GLUT4 protein level expression and fusion into plasma membrane in L6 cells.

### 2.4. AND Promoted Uptake of Glucose in Normal and Insulin-Resistant L6 Cells

We analyzed the effect of AND on the uptake of glucose in L6 cells. Figure 4A shows the chemical structure of andrographolide. As shown in Figure 4B,D, we found that 30, 50, 100 μM AND and 150 μM AND significantly increased the uptake of glucose in both insulin-resistant (IR) and normal L6 cells in a dose-dependent manner. After drug treatment, the survival rate of the cells was determined by an MTT assay. The MTT results showed that both AND and insulin had no toxic effect on L6 cells (Figure 4C).

### 2.5. AND Promoted GLUT4 Fusion into Plasma Membrane and Upregulated the Expression of GLUT4 in L6 Cells

To determine whether AND promoted the uptake of glucose by affecting the glucose regulator GLUT4 in L6 cells, we first carried out immunofluorescence experiments in myc-GLUT4-mOrange-L6 cells. The results showed that both AND and insulin could promote the fusion of GLUT4 into the plasma membrane (Figure 5A). We counted FITC fluorescence on the cell surface to quantify this effect. It was found that in the presence of 100 nM insulin and 30 μM AND, the proportion of FITC fluorescence-positive cells in the GLUT4-mOrange-positive cell number was 76% and 64%, respectively (Figure 5B). To further determine the effect of AND on GLUT4 expression, we explored the expression level of GLUT4 protein in L6 cells after insulin or AND stimulation. The results showed that GLUT4 protein expression was significantly increased in cells treated with 100 nM insulin or different concentrations of AND compared with that in the blank control group (Figure 5C). Therefore, we assumed that AND promoted the expression and translocation of GLUT4 in L6 cells.

### 2.6. AND Activated AMPK and PKC Pathways

Next, we tried to determine the signal transduction pathway involved in AND induced GLUT4 translocation in L6 cells. The cells were treated with different concentrations of AND, and insulin [34], metformin [35], and phorbol 12-myristate 13-acetate (PMA) [36] were used as positive controls for the Akt, AMPK and PKC pathways, respectively. Thees data showed that phosphorylation of the Akt pathway did not change significantly after treatment with different concentrations of AND (Figure 6A), but phosphorylation of AMPK and PKC increased (Figure 6B,C). These results may suggest that AND could activate the AMPK and PKC pathways, but not Akt pathways. 

Subsequently, we used a PI3K/Akt pathway inhibitor (Wortmannin), an AMPK pathway inhibitor (Compound C), and a PKC pathway inhibitor (Gö6983) for further validation. After 30 min of inhibitor treatment, AND was incubated for 30 min, and GLUT4 protein expression level in L6 cells was detected by Western blotting. The results showed that Gö6983, but not Wortmannin and Compound C, inhibited AND-stimulated GLUT4 protein expression (Figure 7). These results suggested that GLUT4 expression induced by AND may be mainly mediated through the PKC pathway.

### 2.7. AND Promoted Ca^2+^ Concentration in L6 Cells

In order to study the role of Ca^2+^ concentration in myc-GLUT4-mOrange-L6 cells stimulated by AND, we first stained the intracellular Ca^2+^ with a Fluo-4 AM fluorescent dye to determine its content in cells. Under confocal laser scanning microscopy, the changes of Ca^2+^ concentration in L6 cells stimulated by AND within 30 min were monitored in real time. The images were taken every 5 min after dosing. We found that intracellular Ca^2+^ levels increased significantly by about 3.4 times after incubation at 30 μM AND for 30 min in a time-dependent manner (Figure 8). Therefore, we speculated that AND could stimulate the increase of Ca^2+^ concentration in L6 cells. However, whether intracellular calcium was involved in AND-stimulated GLUT4 expression and uptake of glucose needs to be further explored.

### 2.8. Ca^2+^ Affected AND-Induced Translocation of GLUT4 and Uptake of Glucose in Myc-GLUT4-mOrange-L6 Cells

To determine whether the increase of intracellular Ca^2 +^ concentration after AND stimulation was related to GLUT4 translocation, we blocked intracellular Ca^2+^ from different sources before 30 μM AND treatment and observed GLUT4 translocation. The changes of FITC green fluorescence in myc-GLUT4-mOrange-L6 cells under 2 mM extracellular Ca^2+^, 0 mM extracellular Ca^2+^, and 0 mM extracellular Ca^2+^ were observed by an immunofluorescence method under the treatment of intracellular Ca^2+^ chelator BAPTA-AM. The results showed that green fluorescence of FITC in L6 cells under three conditions was almost not detected in the control cells. We observed a significant increase in FITC green fluorescence at the cell membrane surface upon stimulation at extracellular Ca^2+^ (2 mM) when 30 μM AND stimulated the cells. Interestingly, under the conditions of 0 mM extracellular Ca^2+^ and 0 mM extracellular Ca^2+^ + 10 μM BAPTA-AM, FITC green fluorescence could hardly be detected on the cell membrane (Figure 9A,B). Therefore, AND-induced GLUT4 fusion into cell membrane was indeed regulated by Ca^2+^ signaling.

Subsequently, we tested the effect of Ca^2+^ concentration on AND-induced uptake of glucose using a glucose oxidase kit. We performed the experiments using a glucose oxidase kit with or without AND or insulin under conditions of 2 mM, 0 mM extracellular Ca^2+^, or 0 mM extracellular Ca^2+^ + 10 μM BAPTA-AM. The results showed that there was no significant difference in the uptake of glucose in the absence of AND or insulin in three conditions. However, when the cells were treated with AND or insulin, the uptake of glucose was significantly lower in the 0 mM extracellular Ca^2+^ concentration treatment than in the 2 mM extracellular Ca^2+^ concentration treatment. Similarly, the uptake of glucose was significantly inhibited with AND or insulin when treated with 0 mM extracellular Ca^2+^ + 10 μM BAPTA-AM (Figure 9C). Therefore, we assumed that AND-induced translocation of GLUT4 and uptake of glucose in L6 cells were indeed affected by Ca^2+^ concentration.

## 3. Discussion

AND is the main ingredient of the natural plant Andrographya paniculata. It is a diterpenoid lactone compound that has heat-removing, detoxifying, anti-inflammatory, and analgesic effects. As a natural antibiotic, it has a special curative effect in bacterial and viral upper respiratory tract infections and diarrhea. Other in vitro studies show that in db/db mice diabetes, AND has a hypoglycemic effect by enhancing intestinal barrier function and increasing microbial diversity [37]. Further studies showed that AND could inhibit the expression of fibronectin in mesangial cells induced by high glucose and inhibit diabetic nephropathy by inhibiting the AP-1 pathway [38]. AND has been used in the treatment of diabetes and its complications; however, there are few studies on the molecular mechanism of its anti-diabetes effect. In this study, we found that AP-EtOAc promoted the uptake of glucose and enhanced GLUT4 transport in L6 cells.AND is probably the major component in AP-EtOAc. AND upregulated GLUT4 expression and promoted GLUT4 fusion into the plasma membrane by activating PKC pathways, while increasing the uptake of glucose in a dose-dependent manner. In addition, AND could increase the content of intracellular Ca^2+^, and the fusion of GLUT4 with plasma membrane and the uptake of glucose induced by AND depended on the existence of Ca^2+^ in L6 cells.

GLUT4 is a member of the GLUT protein family, which is mainly distributed in insulin-sensitive parts, such as skeletal muscle, adipose tissue, and others, and acts in the form of GLUT4 storage vesicles (GSVs). When stimulated by insulin and rising Ca^2+^ concentrations, GLUT4 storage vesicles translocate from the cell to the cell membrane and take up glucose [39]. Therefore, GLUT4 can be used as a therapeutic target for type 2 diabetes. Previous studies reported that AND reduced tumor-necrosis-factor-α-induced insulin resistance in 3T3-L1 adipocytes [40], but its hypoglycemic mechanism in L6 cells is unclear. In this study, we first found that AP-EtOAc promoted the uptake of glucose and enhanced GLUT4 transport in L6 cells (Figure 1). AP-EtOAc increased GLUT4 fusion into the cell membrane (Figure 2A, B) and also enhanced GLUT4 protein expression in L6 cells (Figure 2C). Therefore, we supposed that AP-EtOAc had antidiabetic activity, targeting GLUT4. We further found that AND was the major component in AP-EtOAc (Figure 3). Next, we found that AND could increase the uptake of glucose in insulin-resistant and non-resistant L6 cells through a glucose detection kit (Figure 4B,D), while an MTT assay found that AND had no toxic effect on L6 cells (Figure 4C). Secondly, we also used CLSM to observe the fluorescence distribution in myc-GLUT4-mOrange-L6 cells stimulated by AND through an immunofluorescence experiment. Since the myc tag is inserted into the first outer domain of GLUT4, myc tag is located outside the cell membrane when GLUT4 is fused into the plasma membrane and can adequately serve as a reporter molecule for GLUT4 fusion into the plasma membrane. Meanwhile, AND could induce fusion of GLUT4 into the plasma membrane in myc-GLUT4-mOrange-L6 cells (Figure 5A,B), and the protein expression level also showed that AND promoted the expression of GLUT4 (Figure 5C).

Subsequently, we sought to identify GLUT4-related signal transduction pathways activated by AND. Previous studies showed that GLUT4 translocation and expression involves various signaling pathways, one of the most classical three pathways for the Akt pathway, the AMPK pathway, and the PKC pathway [41,42]. The three pathways can regulate the increase of GLUT4 and the uptake of glucose [43]. When we investigated which of these three signaling pathways were related to AND-induced GLUT4 translocation and expression, we found that AND could significantly activate phosphorylation of the AMPK and PKC pathways but had no effect on the Akt pathway (Figure 6). Meanwhile, when three pathway inhibitors Compound C (AMPK pathway inhibitor), Gö6983 (PKC pathway inhibitor), and Wortmannin (PI3K/Akt pathway inhibitor) were added, Gö6983 significantly inhibited AND-induced expression of GLUT4, while Compound C and Wortmannin had no inhibitory or inducing effect on GLUT4 expression (Figure 7). Therefore, we believe that AND mainly activated the PKC pathway to promote GLUT4 expression.

It has been previously reported that an increased Ca^2+^ concentration can promote GLUT4 translocation to the cell surface and increase glucose transport [44]. In order to further explore the mechanism of AND promoting GLUT4 expression and translocation in L6 cells, we investigated whether Ca^2+^ concentration played a role in this process. First of all, Fluo-4 AM calcium staining showed that AND increased intracellular Ca^2+^ concentration in L6 cells in a time-dependent manner (Figure 8). Secondly, myc-GLUT4-mOrange-L6 cells were stimulated with AND under conditions of 2 mM and 0 mM extracellular Ca^2+^ and 0 mM extracellular Ca^2+^ + 10 μM BAPTA-AM. These results found that AND induced GLUT4 fusion to the plasma membrane in only 2 mM extracellular Ca^2 +^ by immunofluorescence experiments, while under the conditions of 0 mM extracellular Ca^2+^ and 0 mM extracellular Ca^2+^ + 10 μM BAPTA-AM, AND-induced GLUT4 translocation was inhibited (Figure 9A,B). Meanwhile, under the conditions of extracellular Ca^2+^ elimination and intracellular Ca^2+^ presence, AND-induced glucose uptake in L6 cells was significantly reduced (Figure 9C). However, activation of the G-protein-PLC-IP_3_-IP_3_R pathway promotes Ca^2+^ release in the endoplasmic reticulum membrane, resulting in increased intracellular calcium concentrations and thus enhancing the uptake of glucose in L6 cells [26]. Although GLUT4 translocation induced by AND was inhibited at 0 mM extracellular Ca^2+^, there was still a significant enhancomg compared with that with 0 mM extracellular Ca^2+^ + 10 μM BAPTA-AM by immunofluorescence assay (Figure 9B). Therefore, we hypothesized that AND could promote the release of intracellular Ca^2+^ at 0 mM extracellular Ca^2+^, thereby enhancing GLUT4 translocation and the uptake of glucose in L6 cells to a certain extent. In conclusion, AND could increase intracellular Ca^2+^ content, while GLUT4 translocation and uptake of glucose induced by AND depend on Ca^2+^ in L6 cells.

In conclusion, AND promoted the expression of GLUT4 and its fusion into the plasma membrane in L6 cells through PKC pathways, thereby increasing the uptake of glucose, and AND induced both GLUT4 translocation and uptake of glucose in a Ca^2+^-dependent manner (Figure 10). Many animal experiments have shown that AND had certain effects against diabetes and its complications. Our study revealed the molecular mechanism of the anti-diabetes effect of AND and further provided theoretical support for AND as a natural product to treat diabetes.

AND promoted the expression of GLUT4 mainly through the PKC pathways, thereby enhancing the fusion of GLUT4 with the plasma membrane and the uptake of glucose. AND also increased intracellular Ca^2+^ in L6 cells. Moreover, the fusion of GLUT4 into the plasma membrane and the uptake of glucose were induced by AND in Ca^2+^-dependent manner.

## 4. Materials and Methods

### 4.1. Chemicals and Reagents

*Andrographis paniculata (Burm. f)* Nees was collected from Bozhou, Anhui Province, China, and it was identified by Professor DingrongWan at the School of Pharmaceutical Sciences, South-Central Minzu University (SCMZU), Wuhan, China. Andrographolide (98.9%, C_20_H_30_O_5_, M.W.350.45) and metformin were purchased from Shanghai Yuanye Bio-Technology Company (Shanghai, China). GLUT4 (1F8) Mouse mAb (anti-GLUT4, Cat# 2213S), β-Actin (8H10D10) Mouse mAb (anti-β-actin, Cat# 3700S), Phospho-Akt (Ser473) (193H12) Rabbit mAb (anti-p-Akt, Cat# 4058S), Akt Rabbit Ab (Anti-Akt, Cat# 9272S), Phospho-AMPKα (Thr172) (D79.5E) Rabbit mAb (anti-p-AMPK, Cat# 4188S), AMPKα Rabbit Ab (anti-AMPK, Cat# 2532S), and Phospho-PKC (pan) (zeta Thr410) (190D10) Rabbit mAb (anti-p-PKC, Cat# 2060S) were the products of Cell Signaling Technology (Boston, MA, USA). Goat Anti-Rabbit IgG, HRP Conjugated (Cat# CW0103L) and Goat Anti-Mouse IgG, HRP Conjugated (Cat# CW0102L) were purchased from CWBIO (Beijing, China). Anti-c-myc Mouse Monoclonal Antibody (Anti-c-myc, Cat# HT101) and Goat Anti-Mouse IgG (H+L), FITC Conjugate (FITC, Cat# HS211-01) were purchased from Transgen Biotech (Beijing, China). Dorsomorphin (Compound C, Cat# S7840) and GÖ6983 (Cat# S2911) were the products of Selleckchem (Houston, Texas, USA). Wortmannin (Cat# 681676) and BAPTA-AM (Cat# 196419) were purchased from Sigma (Darmstadt, Germany). FBS was purchased from Sijiqing (Hangzhou, China), antibiotics and the minimum were purchased from Gibco (Grand Island, NE, USA). The 2 mM Ca^2+^ in physiological saline solution (PSS) contained the following (in mM): 135 NaCl, 5 KCl, 1 MgCl_2_, 2 CaCl_2_, 10 HEPES, and 10 glucose (pH = 7.4 adjusted with NaOH). The 0 mM Ca^2+^ PSS contained the following (in mM): 135 NaCl, 5 KCl, 1 MgCl_2_, 0.5 EGTA, 10 HEPES, and 10 glucose (pH = 7.4).

### 4.2. Andrographis Paniculata Ethyl Acetate Extract (AP-EtOAc)

The dried andrographis root rhizome was ground into a powder and soaked with an 80% ethanol solution according to the ratio of material-to-liquid 1:5, and the supernatant was collected every 2 days. After suction filtration, it was evaporated to dryness using a rotary evaporator and repeated 3–5 times. Then, the ethanol partial extract was obtained. The ethanol-soluble substance was extracted with ultrapure water and petroleum ether. After the fat was removed, the extract was extracted with ethyl acetate, and the extract was evaporated and dried using a rotary evaporator to obtain the Andrographis extract.

### 4.3. Content Determination

Andrographolide was dissolved with methanol at a concentration of 1.5 mg/mL and filtered with a 0.22 μm microporous membrane filter. A standard solution with a concentration of 1.5 mg/mL was poured into a liquid flask, and 6, 8, 10, 12, 14, 16, 18, and 20 μL was added to each sample. The peak area of andrographolide measured by concentration was plotted as a standard curve. At the same time, AP-EtOAc was dissolved into 6.5 mg/mL with a methanol solution, filtered in the same way as in the above solution, and then put into the liquid bottle. The samples were injected each time under the same conditions. The chromatogram and retention time of andrographolide and AP-EtOAc were compared to determine the existence of andrographolide in AP-EtOAc. The content of andrographolide in AP-EtOAc was determined by substituting the relevant data into the standard curve. The acquisition and processing of data were accomplished by MassLynx™ 4.0 software (Waters, Milford, CT, USA).

### 4.4. Culture and Differentiation of L6 Cells

The L6 cells (Rat myoblasts) were purchased from Procell Life Science & Technology Company (Cat# CL-0136, Wuhan, China) with 10% fetal bovine serum (FBS), 1% of antibiotics (100 U/mL penicillin streptomycin and 100 mu g/mL), and 89% medium of the necessary minimum to prepare the complete medium. L6 cells were cultured at 37 °C in an incubator containing 5% CO_2_ (Thermo Fisher Scientific, New York, NY, USA) and subcultured when the cells’ density was about 80%. In order to obtain differentiated cells, the cells were overgrown at the bottom of the dish, and myotubes were obtained by replacing the differentiation medium with 2% fetal bovine serum (FBS), 1% antibiotics (100 U/mL penicillin and 100 μg/mL streptomycin), and 97% minimum essential medium alpha differentiated for 5–7 days.

### 4.5. Uptake of Glucose Assays of Normal Cells and Insulin Resistant Cells

In this study, glucose uptake levels were measured using a Glucose (GLU-OX) assay kit in the L6 myotubes. First of all, the 96-well plate seeds with approximately 20,000 cells per well and the cells were cultured for proliferation and differentiation. Before starting the experiment, the cells were washed with a PBS buffer and starved with α-MEM for 2 h. After starvation, the previous liquid was sucked out of all the wells, and a serum-free α-MEM group (blank control group), a 100 nM insulin group (positive control group), and four concentration gradient drug groups were designed. The drug in the above configuration was added to 100 μL per well and incubated for 30 min. After incubation, the samples were added to a 96-well plate according to glucose oxidase kit instructions and detected at 505 nm absorbance values using a microplate reader (TECAN, Austria) within 30 min for subsequent calculations. In order to obtain the insulin-resistant cell model, it was necessary to induce the myotubes with a high concentration of an insulin solution. The fully differentiated myotubes were cultured with 1 μM insulin for 24 h, while the negative control cells were cultured in a serum-free α-MEM medium. After obtaining the insulin resistance model, the experimental procedure was the same as with the uptake of glucose experiment described above.

### 4.6. MTT Colorimetric Assay for Cell Survival Rate

The L6 cells were inoculated on 96-well plates for culture differentiation, and the experiment was conducted after 5–7 d of differentiation into L6 myotubes. First, the cells were starved, and the operation was the same as with the uptake of glucose experiment. The liquid in the 96-well plates was discarded after 30 min of dose-adding reaction, and 100 μL of MTT (0.5 mg/mL) solution was added to each well. The MTT of L6 cells after 4 h incubation was aspirated, and a DMSO solution at 150 μL per well was added; after shaking for 60 s, the absorbance value at 492 nm was measured by a microplate reader for subsequent calculation.

### 4.7. Fusion Analysis of GLUT4 with Plasma Membrane

L6 cells were transfected with a lentivirus vector GV348-myc-GLUT4-mOrange, which encodes an mOrange fusion protein of GLUT4 tagged with myc epitopes [45]. The myc-GLUT4-mOrange-L6 cells were cultured on round glass slides in 6-well plates until they differentiated into L6 myotubes. The differentiated cells were starved with free serum medium for 2 h. The blank control group, 100 nM insulin group (positive control group), and 60 μg/mL AP-EtOAc and 30 μM AND group (experimental group) were set up in 6-well plates. After 30 min of drug administration, the cells were immobilized with 3% paraformaldehyde, incubated with 50 mM glycine for 20 min to remove background impurities, and then sealed with 2% bovine serum albumin (BSA) in PBS. After blocking, the cells were incubated with anti-c-myc for 1 h at room temperature. Then, the cells were washed three times with 2% BSA in PBS and incubated with a secondary antibody (goat anti-mouse-FITC) against light. The cells were washed 3 times with BSA and then washed 3 times in PBS before placing the coverslip upside down on the glass slide. Finally, the intensity and distribution of red and green fluorescence in the cells were observed by a Confocal laser scanning microscope (LSM 700; Carl Zeiss, Jena, Germany), and then the expression of GLUT4 in the cells and the fusion of GLUT4 with the plasma membrane were analyzed. The images were analyzed by Zen 2010 Software (Carl Zeiss, Jena, Germany), and the number of cells carrying FITC green fluorescence was counted.

### 4.8. Preparation of L6 Cells Protein

The differentiated L6 cells were starved in free serum medium for 2 h, and then the corresponding concentration of drugs was added and incubated in an incubator for 30 min. After the completion of drug stimulation, the cells were immediately taken out and placed on ice and then washed with precooled PBS at 4 °C for 3 times. At the same time, 100 μL of RIPA lysis solution containing PMSF and phosphatase inhibitor was added to each dish. The cells in the dish were collected in 1.5 mL EP tubes and then extracted and broken for 30 times with 27-aperture and 12-aperture syringes successively. The whole operation was carried out on ice. Then, the lysate was centrifuged at 12,000 rpm at 4 °C for 15 min; the supernatant was total cell protein. The concentration of protein was detected by a BCA protein concentration determination kit (Beyotime, Shanghai, China), and the absorbance value at 562 nm was detected by a microplate reader. All protein samples were denatured at 65 °C for 10 min after being added into the protein loading buffer and stored in the refrigerator at −20 °C.

### 4.9. Western Blot Analysis

According to the PAGE Gel rapid preparation Kit (10%), the separation gel and the concentrated gel were prepared. After the protein sample was run for 2 h, the protein was transferred to a nitrocellulose filter membrane (NC) and sealed with 5% BSA in PBS with Tween-20 (PBST) at room temperature for 2 h. A diluted primary antibody was prepared with a PBST buffer and a corresponding primary antibody in a ratio of 1000:1 and incubated overnight at 4 °C in a shaker. The next day, the corresponding horseradish peroxidase conjugated secondary antibody (1:10,000) was incubated for 1 h at room temperature. The gray value of protein was quantitatively determined by a ChemiDoc XRS system (Bio-RAD, Hercules, CA, USA).

### 4.10. Intracellular Ca^2+^ Assays

The myc-GLUT4-mOrange-L6 cells were cultured on a cover glass until they differentiated into myotube cells and then starved in serum-free α-MEM for 2 h. Next, the cells were treated with a PSS solution containing 2.5 μM Fluo4-AM, incubated at room temperature for 20 min and washed twice with PSS. Then, the fluorescence intensity changes of Fluo4-AM in cells within 30 min after the addition of AND were detected by a confocal laser scanning microscope at 488 nm. Images were taken 10 s before AND treatment and every 5 min after AND treatment. Zen 2010 software was used to analyze and record the changes of cytoplasmic Fluo-4 fluorescence.

### 4.11. Data Analysis

To determine whether there were significant differences between groups, we performed t-tests by using GraphPad Prism 8.0 software (GraphPad Software, Lung cancer San Diego, CA, USA). Meanwhile, multivariate analysis was performed by two-way ANOVA. The data are shown as the mean ± standard error. The *n* values represent the number of experiments repeated. The difference was considered statistically significant if *p* < 0.05.

## 5. Conclusions

In this study, we found that AP-EtOAc had anti-diabetic activity, and AND was the major component in AP-EtOAc. AND promoted the expression of GLUT4 and its fusion with the plasma membrane in L6 cells through PKC pathways in a Ca^2+^-dependent manner, so as to increase uptake of glucose.

## Figures and Tables

**Figure 1 pharmaceuticals-15-01346-f001:**
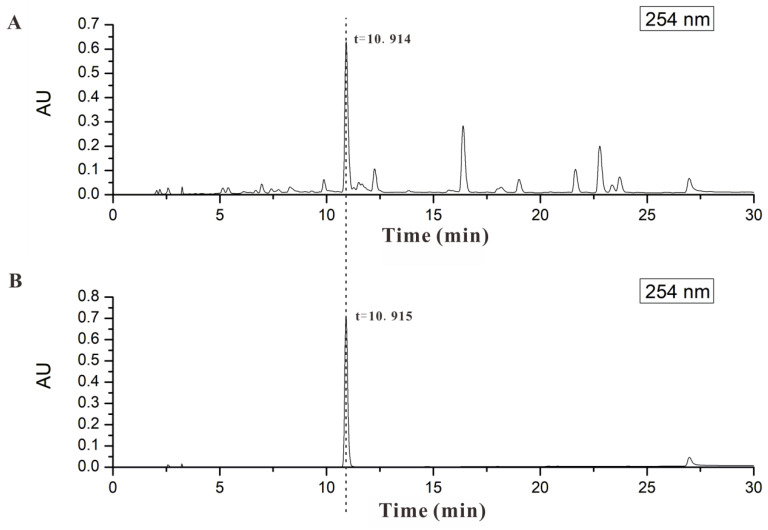
AND was the major component in AP-EtOAc. (**A**) HPLC analysis of AP-EtOAc shown at 254 nm. (**B**) HPLC analysis of AND shown at 254 nm.

**Figure 2 pharmaceuticals-15-01346-f002:**
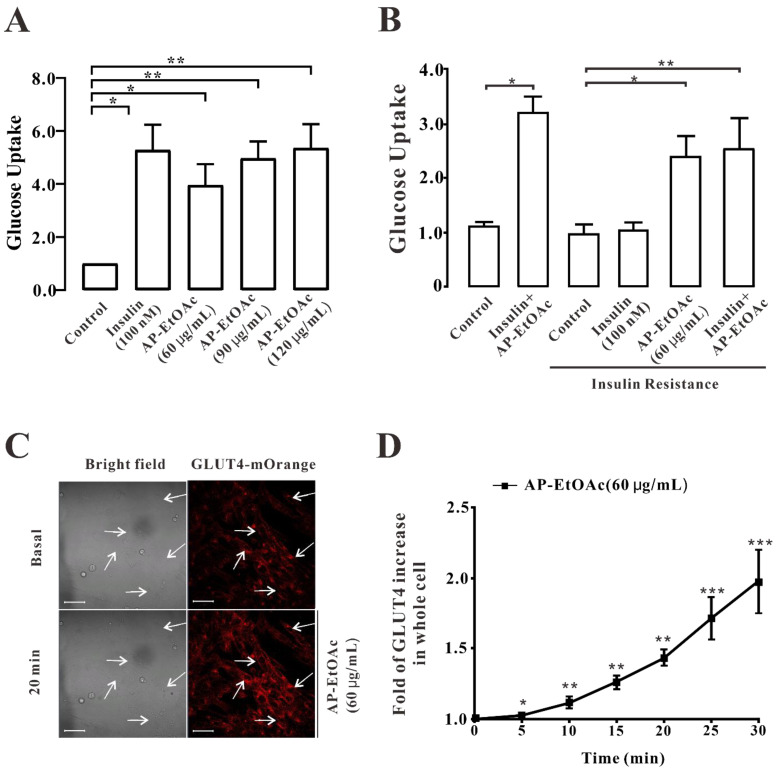
AP-EtOAc promoted the uptake of glucose and enhanced GLUT4 transport in L6 cells. (**A**) Values on the Y-axis represent the fold relationship of glucose uptake by L6 cells compared with that in normal conditions. Uptake of glucose detected using a Glucose oxidase kit in L6 cells, the data represent the mean ± s.e.m. of values from three separate experiments. The control group was considered as 1 for data analysis. (**B**) Values on the Y-axis represent the fold relationship of glucose uptake by L6 cells compared with that in normal conditions. Uptake of glucose in L6 cells induced by AP-EtOAc in insulin resistant L6 cells, the data represent the mean ± s.e.m. of values from three separate experiments. (**C**) Images of 60 μg/mL AP-EtOAc stimulating GLUT4 transport in L6 cells. Scale bar = 50 μm. (**D**) Calculation of fluorescence intensity in myc-GLUT4-mOrange-L6 cells using Zen 2010 software, *n* = 30 cells. *: *p* < 0.05; **: *p* < 0.01, ***: *p* < 0.001.

**Figure 3 pharmaceuticals-15-01346-f003:**
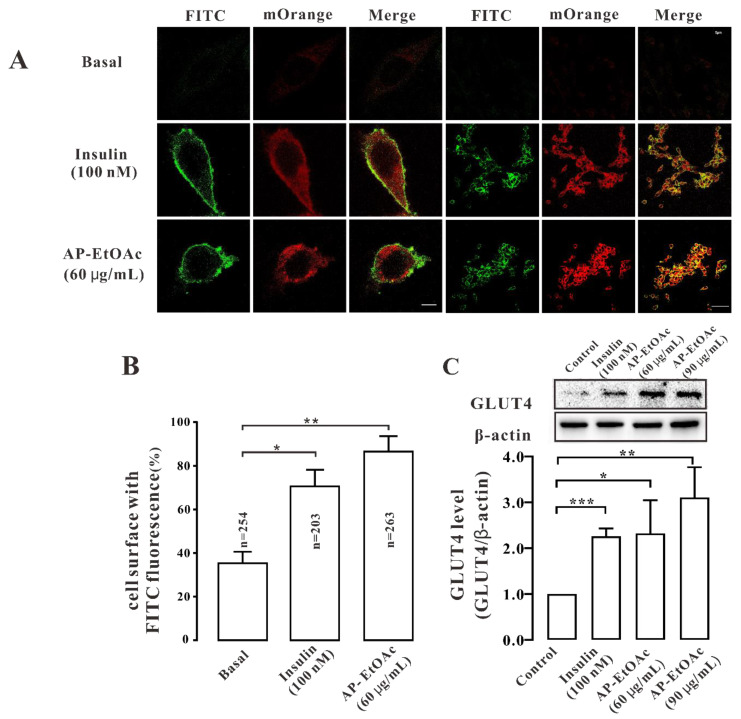
AP-EtOAc enhanced GLUT4 protein expression and GLUT4 fusion with the plasma membrane in L6 cells. (**A**) FITC fluorescence assay in myc-GLUT4-mOrange cells treated with 60μg/mL AP-EtOAc or 100 nM insulin. Scale bar: 5 μm in single cell image, 50 μm in multi-cell image. (**B**) Number of GLUT4-mOrange-positive cells, the data represent the mean ± s.e.m. of values from three separate experiments with between 200 and 300 cells being examined in each experiment. (**C**) The expression level of GLUT4 after treated with 100 nM insulin or different concentrations of AP-EtOAc for 30 min in L6 cells. The control group was considered as 1 for data analysis. The data represent the mean ± s.e.m. of values from three separate experiment. *: *p* < 0.05; **: *p* < 0.01, ***: *p* < 0.001.

**Figure 4 pharmaceuticals-15-01346-f004:**
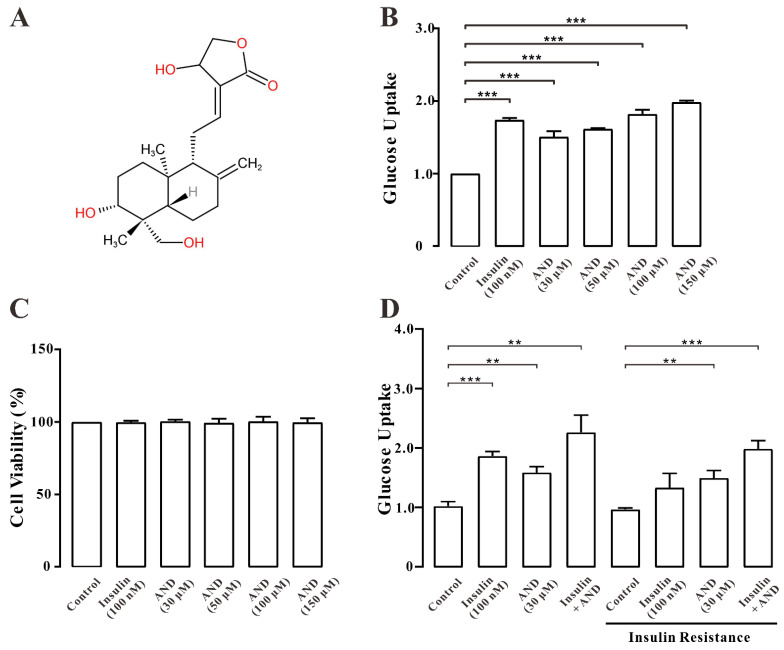
AND promoted the uptake of glucose in L6 cells. (**A**) Chemical structure of AND. (**B**) Values on the Y-axis represent the fold relationship of glucose uptake by L6 cells compared with that in normal conditions. Uptake of glucose in L6 cells under the incubation of AND or insulin, the data represent the mean ± s.e.m. of values from three separate experiments. (**C**) Toxicity of AND to L6 cells, the data represent the mean ± s.e.m. of values from three separate experiments. (**D**) Values on the Y-axis represent the fold relationship of glucose uptake by L6 cells compared with that in normal conditions. Uptake of glucose in L6 cells induced by AND in insulin-resistant L6 cells, the data represent the mean ± s.e.m. of values from three separate experiments. **: *p* < 0.01; ***: *p* < 0.001.

**Figure 5 pharmaceuticals-15-01346-f005:**
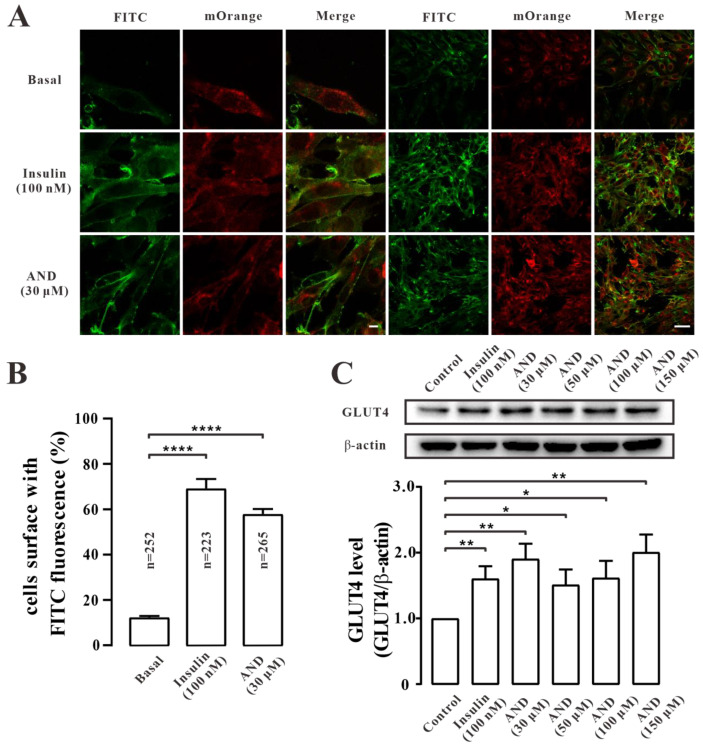
AND promoted GLUT4 expression and fusion into plasma membrane. (**A**) FITC fluorescence analysis of L6 cells treated with AND or Insulin. Scale bar: 10 μm for single-cell images and 50 μm for multi-cell images. (**B**) The number of GLUT4-mOrange-positive cells, the data represent the mean ± s.e.m. of values from three separate experiments with between 200 and 300 cells examined in each experiment. (**C**) The expression level of GLUT4 after treatment with 100 nM insulin or different concentrations of AND for 30 min in L6 cells. The control group was considered as 1 for data analysis. The data represent the mean ± s.e.m. of values from three separate experiments. *: *p* < 0.05; **: *p* < 0.01; ****: *p* < 0.0001.

**Figure 6 pharmaceuticals-15-01346-f006:**
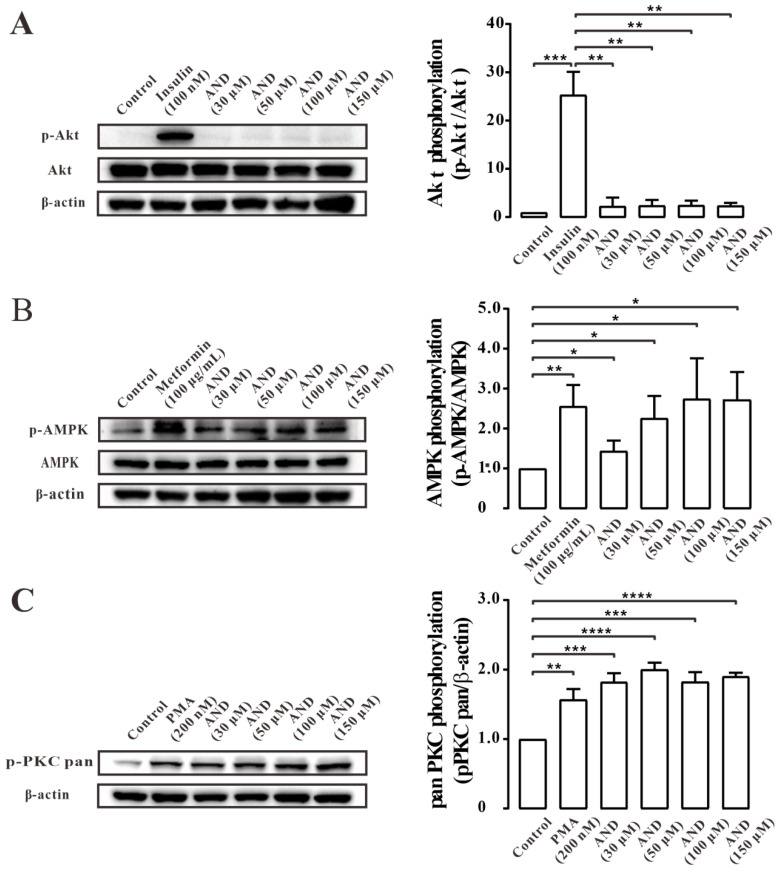
AND induced phosphorylation of PKC and AMPK signaling pathways. (**A**) The phosphorylation of Akt in L6 cells treatment with insulin and different concentrations of AND. The control group was considered as 1 for data analysis. The data represent the mean ± s.e.m. of values from three separate experiments. (**B**) The phosphorylation of AMPK in L6 cells treatment with metformin and different concentrations of AND. The control group was considered as 1 for data analysis. The data represent the mean ± s.e.m. of values from three separate experiments. (**C**) The phosphorylation of PKC in L6 cells treatment with PMA and different concentrations of AND. The control group was considered as 1 for data analysis. The data represent the mean ± s.e.m. of values from three separate experiments. *: *p* < 0.05; **: *p* < 0.01; ***: *p* < 0.001; ****: *p* < 0.0001.

**Figure 7 pharmaceuticals-15-01346-f007:**
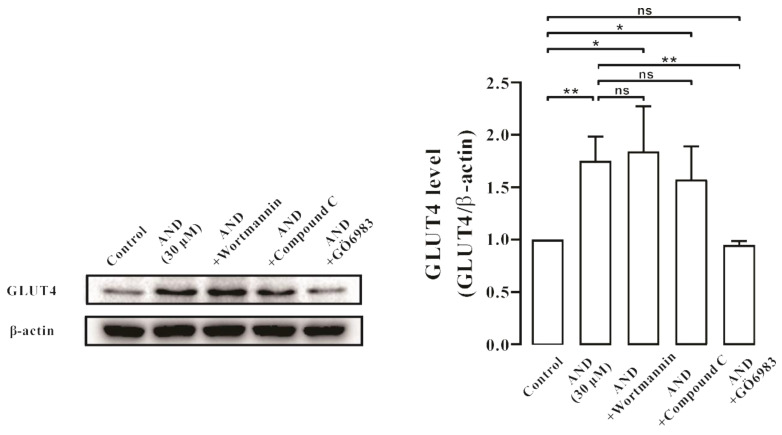
AND induced GLUT4 expression through the PKC signaling pathway. The control group was considered as 1 for data analysis. The expression of GLUT4 induced by AND under the action of three inhibitors. ns: *p* >0.05; *: *p* < 0.05; **: *p* < 0.01.

**Figure 8 pharmaceuticals-15-01346-f008:**
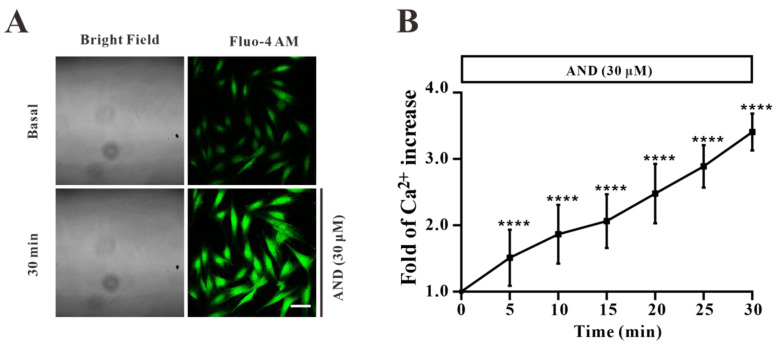
AND induced increased Ca^2+^ concentration in L6 cells. (**A**) Images of Ca^2+^ concentration changes in L6 cells treated with AND. Scale bar: 50 μm. (**B**) The change of intracellular Ca^2+^ fluorescence intensity caused by AND within 30 min, *n* = 15 cells. ****: *p* < 0.0001.

**Figure 9 pharmaceuticals-15-01346-f009:**
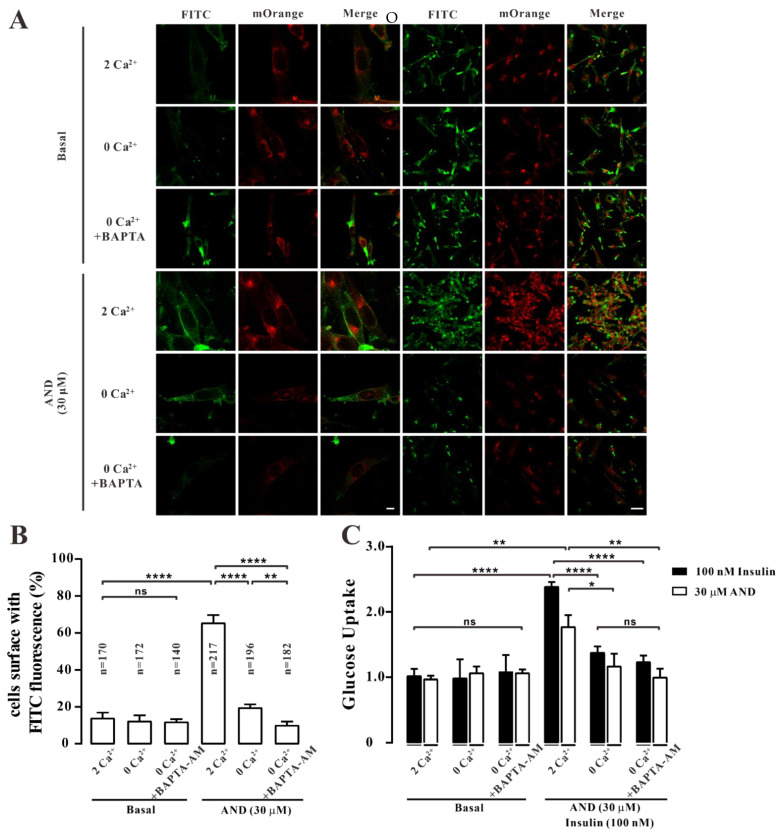
Translocation of GLUT4 and uptake of glucose induced by AND in L6 cells blocked by 0 mM extracellular Ca^2+^ and 0 mM extracellular Ca^2+^ + 10 μM BAPTA-AM. (**A**) L6 cells were stimulated by AND under the conditions of 2 mM extracellular Ca^2+^, 0 mM extracellular Ca^2+^, and 0 mM extracellular Ca^2+^ + 10 μM BAPTA-AM. The distribution of red and green fluorescence in L6 cells was detected by confocal laser scanning microscopy. Scale bar: 10 μm for single-cell images and 50 μm for multi-cell images. (**B**) The number of GLUT4-mOrange-positive cells data represent the mean ± s.e.m. of values from three separate experiments, with between 200 and 300 cells examined in each experiment. (**C**) Uptake of glucose induced by insulin or AND in the three Ca^2+^ buffering systems. The three groups on the left are the control group, and the three groups on the right represent glucose uptake after adding insulin and AND. The data represent the mean ± s.e.m. of values from three separate experiments. ns: *p* > 0.05; *: *p* < 0.05; **: *p* < 0.01; ****: *p* < 0.0001.

**Figure 10 pharmaceuticals-15-01346-f010:**
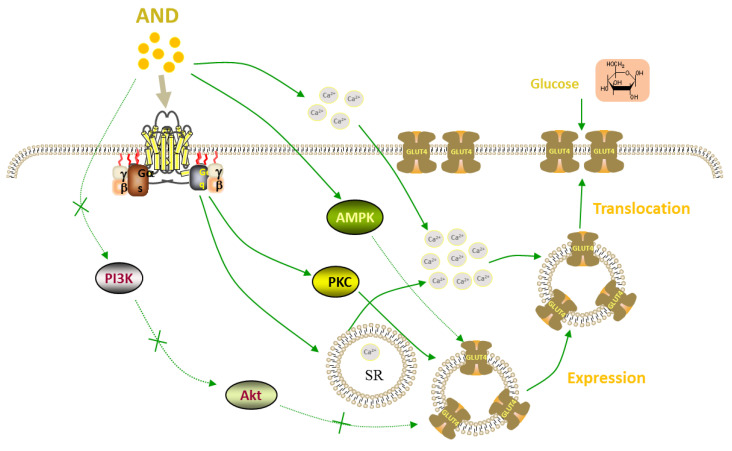
Proposed model of the AND-induced increase of glucose uptake in L6 cells.

## Data Availability

Data is contained within the article.

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
