# Peer review of "Andrographolide Promotes Uptake of Glucose and GLUT4 Transport through the PKC Pathway in L6 Cells"

_pharmaceuticals, 2022, doi:10.3390/ph15111346_

Round 1
Reviewer 1 Report
The keywords should appear in alphabetical order.
In line 176, margin is not correct.
The numbers of the figures are not well placed
Author Response
The keywords should appear in alphabetical order.
Response: The alphabetical order of the keywords shows that line 30 has been modified in the manuscript.
In line 176, margin is not correct.
Response: The margin display has been corrected in line 153 of the manuscript.
The numbers of the figures are not well placed
Response: The position of Figure 4 has been corrected on line 232 in the manuscript.
Reviewer 2 Report
In the present study entitled “Andrographolide Promotes Uptake of Glucose and GLUT4 Transport Through the PKC Pathway in L6 cells,” the authors have shown the promoting effect of Andrographolide on the glucose uptake, GLUT4 expression and fusion with the plasma membrane and they examined the signalling pathways regulating by Andrographolide and the effect of Ca concentration on the action of Andrographolide. The experiments and the results sound interesting, however, there are a few issues that should be addressed:
Minor comments
Introduction, line 55: streptozotocin is used twice in the sentence.
Page 3, line 77: Figure 4. is incorporated into the materials and methods section.
Figure 4. What is the value of glucose uptake on the Y-axis?
Figure 4. C, D it is unnecessary to sign if no significant alteration was revealed in the experiment.
Lines 93,95 both u and µ were used by the authors.
Please give the manufacturer of FBS, antibiotics and culture medium.
It is not clear what was the differentiation medium for L6 cells and what 2% FBS was used for.
It is very difficult to follow the experimental protocol for L6 cells in the 2.5 section. Please rewrite this section for a better understanding.
What was the cell density in the 96 well plates?
Why the authors used 492 nm for the measurement of the MTT assay? Usually, the wavelength to measure the absorbance of the formazan product is between 500 and 570 nm (or it can be measured with a filter in the wavelength range of 550 - 600 nm).
In the 2. 9. section many tiny mistakes should be corrected.
Line 164. Please give the origin of the cell line or give a reference for the construction of this cell line.
Line 165. Please give the name of PSS at first mention.
Line 174. “The difference was statistically significant (p < 0.05)”. The difference was considered statistically significant if the p < 0.05.
Line 179. “As shown in Figure 1A and Figure 1B, AP-EtOAc is capable of uptake of glucose…” I suppose that AP-EtOAc is capable of increasing/triggering the uptake of glucose.
Line 190-192. “from three separate experiment” from three separate experiments.
Figure 1. What is the value of glucose uptake on the Y-axis?
Major comments
Why did the Authors use only 60 µg/mL of the drug in the insulin resistant cells if the 90 µg/mL was comparable with the effect of insulin in normal L6 cells?
Figure 1C The authors should examine the effect of insulin on the GLUT4 translocation
as well, as a positive control for the comparison of the effectiveness between insulin and AP-EtOAc.
Figure 2C. The quality of both GLUT4 and β-actin is very poor and seemed to be modified by the huge modification of the contrast and brightness. I suppose that the Authors could not analyse properly this WB image, based on the expression of β-actin I do not know how the Authors calculated that the 90 ug/mL AP-EtOAc increased the GLUT4 level. Better WB should be added to the manuscript.
Anyway, if the authors write a 0.9-fold increase without mentioning that the control level was around 0.5, it is difficult to agree with the elevation. But, if the Authors compare the protein levels, there is a 2-fold increase in the GLUT4 protein level.
The composition analysis of the extract should be placed at the beginning of the results. Can the Authors give the exact concentration (µM) of the AND in the extract? It would be useful for comparing it with the AND experiments. What are the other components of the extract seen on the chromatogram?
Figure 5-7. The WBs are oversaturated. It is difficult to compare the expression levels this way. In the case of Akt, the oversaturation hides that the Authors revealed double bands on the blot. It is useless because the antibody that was used in the experiments generally reveals a double band for Akt. In the case of p-AMPK, it would be useful to see the original WB. It seems that this is a technical problem. Since the Authors used the ChemiDoc XRS system for the examination of protein bands it would be easy to make an accurate setting for proper visualisation. Please, do not sing the nonsignificant alterations in the figures.
In figure 2, the Authors applied S.E.M. and the original value for the control. Later the authors considered the control as 1 without an error bar. Please unify the figures and mention in the figure legends if control was regarded as 1.
The interpretation of the results in the 3.8 section is ambiguous. Maybe due to the mistakes in the description of the experiments e.g. “…cell membrane surface at 2 mM extracellular Ca2+ when 30 uM AND stimulated cells.”; “…under the conditions of 0 mM and 0 mM extracellular Ca2+ + 10 uM BAPTA-AM.”; “…with or without AND or insulin under conditions of 2 mM, 0 mM extracellular Ca2+, or 0 mM extracellular Ca2+ + 10 uM BAPTA-AM.” Please correct these sentences.
Why the authors used the extract in the first three experiments and turned to AND later? To prove the effectiveness of the AP-EtOAc, the Authors should compare the effects of AND and AP-EtOAc.
For the statistical analysis, the Authors should use two-way ANOVA, especially when there are two variable factors and three experimental groups (e.g. Fig 9).
In Figure 9C what is the difference between the first three and the second three groups?
Author Response
Minor comments
Introduction, line 55: streptozotocin is used twice in the sentence.
Response: It has been corrected in line 55 of the introduction.
Page 3, line 77: Figure 4. is incorporated into the materials and methods section.
Response: The position of Figure 4 has been corrected on line 232 in the manuscript.
Figure 4. What is the value of glucose uptake on the Y-axis?
Response: Values on the Y-axis represent the fold relationship of glucose uptake by L6 cells compared with normal conditions. The lines 199、201、234 and 237 have been added in the manuscript.
Figure 4. C, D it is unnecessary to sign if no significant alteration was revealed in the experiment.
Response: The sign with no significant differences in Figure 4C and Figure 4D have been removed in the manuscript.
Lines 93,95 both u and µ were used by the authors.
Response: It has been revised in lines 90 and 92 of the manuscript.
Please give the manufacturer of FBS, antibiotics and culture medium.
Response: The manufacturers of FBS, antibiotics, and media have been added in line 77 of the manuscript.
It is not clear what was the differentiation medium for L6 cells and what 2% FBS was used for.
Response: In order to obtain differentiated cells, the cells were overgrown at the bottom of the dish and myotubes were obtained by replacing differentiation medium with 2% fetal bovine serum (FBS), 1% antibiotics (100 U/mL penicillin and 100 μg/mL streptomycin) and 97% minimum essential medium alpha differentiated for 5-7 days. The configuration of 2% differentiation medium and the action of 104 lines in the manuscript were supplemented.
It is very difficult to follow the experimental protocol for L6 cells in the 2.5 section. Please rewrite this section for a better understanding.
Response: The 109 lines in the manuscript have been revised.
What was the cell density in the 96 well plates?
Response: The 96-well plate seeds approximately 20,000 cells per well, and the cells are cultured for proliferation and differentiation. The cell densities in 96-well plates were modified at 109 lines in the manuscript.
Why the authors used 492 nm for the measurement of the MTT assay? Usually, the wavelength to measure the absorbance of the formazan product is between 500 and 570 nm (or it can be measured with a filter in the wavelength range of 550 -600 nm).
Response: MTT is a yellow compound, which is a dye that accepts hydrogen ions and can act on the respiratory chain in the mitochondria of living cells. Under the action of succinate dehydrogenase and cytochrome C, exogenous MT is reduced to water insoluble blue purple crystalline methylzan (Formazan) and deposited in cells, while dead cells have no such function. Dimethyl sulfoxide (DMSO) can dissolve methylzan in cells, and its light absorption value can be measured at 490nm wavelength with an enzyme-linked immunosorbent assay, which can indirectly reflect the number of living cells. Within a certain range of cells, the amount of MT crystal formation is proportional to the number of living cells.
In the 2. 9. section many tiny mistakes should be corrected.
Response: The 156 lines (section 2.9) in the manuscript have been corrected.
Line 164. Please give the origin of the cell line or give a reference for the construction of this cell line.
Response:
The L6 cells were transfected with the lentivirus vector GV348-myc-GLUT4-mOrange, which encodes a mOrange fusion protein of GLUT4 tagged with myc epitopes [1]. The source of cell line construction is added to the manuscript at line 130, and the reference is attached to line 493 of the manuscript.
Reference:
- Zhao, P.; Ming, Q.; Qiu, J.; Tian, D.; Liu, J.; Shen, J.; Liu, Q.H.; Yang, X. Ethanolic Extract of Folium Sennae Mediates the Glucose Uptake of L6 Cells by GLUT4 and Ca(2). Molecules (Basel, Switzerland) 2018, 23, doi:10.3390/molecules23112934.
Line 165. Please give the name of PSS at first mention.
Response: The composition of PSS has been supplemented in 78 lines of the manuscript.
Line 174. “The difference was statistically significant (p < 0.05)”. The difference was considered statistically significant if the p < 0.05.
Response: The difference was considered statistically significant if the p < 0.05. The 173 lines in the manuscript have been corrected.
Line 179. “As shown in Figure 2A and Figure 2B, AP-EtOAc is capable of uptake of glucose…” I suppose that AP-EtOAc is capable of increasing/triggering the uptake of glucose.
Response:
We supposed that AP EtOAc can increase the uptake of glucose in both insulin resistance (IR) and non-insulin resistance L6 cells, while 100 nM insulin only induces the increase of glucose uptake of non-insulin resistance cells, as shown in Figure 2A and Figure 2B. The 188 lines in the manuscript have been corrected.
Line 190-192. “from three separate experiment” from three separate experiments.
Response: The lines 200 and 203 have been revised in the manuscript.
Figure 1. What is the value of glucose uptake on the Y-axis?
Response: Values on the Y-axis represent the fold relationship of glucose uptake by L6 cells compared with normal conditions. The lines 199 and 201 have been added in the manuscript.
Major comments
Why did the Authors use only 60 µg/mL of the drug in the insulin resistant cells if the 90 µg/mL was comparable with the effect of insulin in normal L6 cells?
Response: This is because 60 µg/mL of the drug can significantly enhance the uptake of glucose in L6 cells. The purpose of using the insulin resistance model to carry out experiments is to prove whether the drug can enable cells with insulin resistance to still take glucose, thus improving the insulin resistance of cells. Therefore, it is enough to use the drug concentration of 60 µg/mL.
Figure 1C The authors should examine the effect of insulin on the GLUT4 translocation. As well, as a positive control for the comparison of the effectiveness between insulin and AP-EtOAc.
Response:
Insulin and AP-EtOAc stimulate GLUT4 translocation to plasma membrane with immunofluorescence data, as shown in Figure 3A and 3B. Although one experiment is real-time monitoring and the other is fixed cells imaging, the cells used in both experiments are myc-GLUT4-mOrange L6 cells, and the microscope used is laser confocal microscope. Therefore, it can be used as a positive control for comparing the efficacy of insulin and AP-EtOAc.
Figure 2C. The quality of both GLUT4 and β-actin is very poor and seemed to be modified by the huge modification of the contrast and brightness. I suppose that the Authors could not analyse properly this WB image, based on the expression of β-actin I do not know how the Authors calculated that the 90 ug/mL AP-EtOAc increased the GLUT4 level. Better WB should be added to the manuscript.
Response: The quality of GLUT4 and β- actin protein has been corrected, as shown in Figure 3C. The original data is also provided in Figure 1 of the supplementary material.
Anyway, if the authors write a 0.9-fold increase without mentioning that the control level was around 0.5, it is difficult to agree with the elevation. But, if the Authors l. compare the protein levels, there is a 2-fold increase in the GLUT4 protein level
Response: The controls in Figures 2A, B, and 3C of the manuscript are considered as 1 and are mentioned in the figures legend.
The composition analysis of the extract should be placed at the beginning of the results. Can the Authors give the exact concentration (µM) of the AND in the extract? It would be useful for comparing it with the AND experiments. What are the other components of the extract seen on the chromatogram?
Response: The component analysis of the extract has been placed at the beginning of the results, as shown in Figure 1. In the liquid phase experiment, the concentration of AND standard is 1.5 mg/mL, and the concentration of mixture is 6.5 mg/mL. By setting standard samples of different volumes, the corresponding data were obtained and the standard curve was calculated. It was found that the content of AND in the mixture was about 22.71%. The components of other extracts need to be further purified and identified.
Figure 5-7. The WBs are oversaturated. It is difficult to compare the expression levels this way. In the case of Akt, the oversaturation hides that the Authors revealed double bands on the blot. It is useless because the antibody that was used in the experiments generally reveals a double band for Akt. In the case of p-AMPK, it would be useful to see the original WB. It seems that this is a technical problem. Since the Authors used the ChemiDoc XRS system for the examination of protein bands it would be easy to make an accurate setting for proper visualisation. Please, do not sing the nonsignificant alterations in the figures.
Response: The original data maps related to protein immunoblotting have been placed in the supplementary materials. Figures 3C, 5C, 6A, 6B, 6C and 7 correspond to Supplementary Figures 1, 2, 3, 4, 5 and 6.
In figure 2, the Authors applied S.E.M. and the original value for the control. Later the authors considered the control as 1 without an error bar. Please unify the figures and mention in the figure legends if control was regarded as 1.
Response: The control group was considered as 1 for data analysis and has been supplemented in line 201, 223, 257, 270, 272, 274, 283 of the manuscript.
The interpretation of the results in the 3.8 section is ambiguous. Maybe due to the mistakes in the description of the experiments e.g. “…cell membrane surface at 2 mM extracellular Ca2+ when 30 uM AND stimulated cells.”; “…under the conditions of 0 mM and 0 mM extracellular Ca2+ + 10 uM BAPTA-AM.”; “…with or without AND or insulin under conditions of 2 mM, 0 mM extracellular Ca2+, or 0 mM extracellular Ca2+ + 10 uM BAPTA-AM.” Please correct these sentences.
Response: The changes of FITC green fluorescence in myc-GLUT4-mOrange-L6 cells under 2 mM extracellular Ca2+, 0 mM extracellular Ca2+ and 0 mM extracellular Ca2+ were observed by immunofluorescence method under the treatment of intracellular Ca2+ chelator BAPTA-AM. The results showed that the green fluorescence of FITC in L6 cells under three conditions was almost not detected in the control cells. We observed a significant increase in FITC green fluorescence at the cell membrane surface upon stimulation at extracellular Ca2+ (2 mM) when 30 mM AND stimulated cells. Interestingly, under the conditions of 0 mM extracellular Ca2+and 0 mM extracellular Ca2+ + 10 mM BAPTA-AM, FITC green fluorescence could hardly be detected on the cell membrane (Figure 9A, B). Therefore, AND induced GLUT4 fusion into cell membrane is indeed regulated by Ca2+ signaling. It has been revised in line 303 of the manuscript.
Why the authors used the extract in the first three experiments and turned to AND later? To prove the effectiveness of the AP-EtOAc, the Authors should compare the effects of AND and AP-EtOAc.
Response: AP-EtOAc was used to prove that the ethyl acetate extract of Andrographis Paniculata has an anti-diabetes effect in the extraction mixture of Andrographis Paniculata, and we also proved through experiments that AP-EtOAc can promote glucose uptake by L6 cells and the expression of GLUT4. Later, we wanted to know which compounds in the mixture played the main hypoglycemic role, so we used AND. Our aim is to find out the effective compounds in the mixture that have hypoglycemic effects, rather than in-depth study of AP-EtOAc.
For the statistical analysis, the Authors should use two-way ANOVA, especially when there are two variable factors and three experimental groups (e.g. Fig 9).
Response: For a single variable, we mainly use t-tests for analysis, while for multivariate analysis, we used two-way ANOVA, which is omitted in the article and only written t-tests. The 171 lines in the manuscript have been added.
In Figure 9C what is the difference between the first three and the second three groups?
Response: The three groups on the left of Figure 9C are the control group, and the three groups on the right represent the glucose uptake after adding insulin and AND. This has been supplemented in Figure 9C of the manuscript with a corresponding note added at line 327.

Round 2
Reviewer 2 Report
The Authors replied to my concerns acceptably.